# Poly(l-lactide-*co*-caprolactone-*co*-glycolide)-Based Nanoparticles as Delivery Platform: Effect of the Surfactants on Characteristics and Delivery Efficiency

**DOI:** 10.3390/nano12091550

**Published:** 2022-05-03

**Authors:** Magda M. Rebanda, Simona Bettini, Laura Blasi, Antonio Gaballo, Andrea Ragusa, Alessandra Quarta, Clara Piccirillo

**Affiliations:** 1CNR Nanotec, Institute of Nanotechnology, Campus Ecotekne, Via Monteroni, 73100 Lecce, Italy; magda.rebanda@hotmail.com (M.M.R.); laura.blasi@cnr.it (L.B.); antonio.gaballo@nanotec.cnr.it (A.G.); andrea.ragusa@unisalento.it (A.R.); 2Laboratório Associado, CBQF—Centro de Biotecnologia e Química Fina, Escola Superior de Biotecnologia, Universidade Católica Portuguesa, 4169-005 Porto, Portugal; 3Department of Biological and Environmental Sciences and Technologies, University of Salento, Via Monteroni, 73100 Lecce, Italy; simona.bettini@unisalento.it; 4Institute for Microelectronics and Microsystems, Campus Ecotekne, Via Monteroni, 73100 Lecce, Italy

**Keywords:** polymeric nanoparticle, surfactant-drug interaction, doxorubicin, SN-38

## Abstract

Polymeric nanoparticles made of the copolymer Poly(L-lactide-*co*-caprolactone-*co*-glycolide) were prepared using the solvent evaporation method. Two different surfactants, polyvinyl alcohol and dextran, and a mixture of the two were employed. The three types of nanoparticles were used as hosting carriers of two chemotherapeutic drugs, the hydrophilic doxorubicin and the hydrophobic SN-38. The morphostructural characterization showed similar features for the three types of nanoparticles, while the drug encapsulation efficiency indicated that the dextran-based systems are the most effective with both drugs. Cellular studies with breast cancer cells were performed to compare the delivery capability and the cytotoxicity profile of the three nanosystems. The results show that the unloaded nanoparticles are highly biocompatible at the administered concentrations and confirmed that dextran-coated nanoparticles are the most efficient vectors to release the two drugs, exerting cytotoxic activity. PVA, on the other hand, shows limited drug release in vitro, probably due to strong interactions with both drugs. Data also show the release is more efficient for doxorubicin than for SN-38; indeed, the doxorubicin IC_50_ value for the dextran-coated nanoparticles was about 35% lower than the free drug. This indicates that these nanocarriers are suitable candidates to deliver hydrophilic drugs while needing further modification to host hydrophobic molecules.

## 1. Introduction

In the last two decades, advances in nanotechnology boosted the development of colloidal formulations of nanoparticles which have been considered for biomedical use [1,2]. Among the different applications, drug delivery is one of the most studied [3], especially for cancer treatment [2,4], as nanoformulation can improve the pharmacokinetics and the pharmacodynamics of the drug. Indeed, NPs can be employed as Drug Delivery Systems (DDSs) by encapsulating selected anti-cancer drugs and/or having them bonded/adsorbed on their surface [5]. In this way, the DDSs can also reduce the adverse side effects normally experienced with systemic drug treatment, the non-specific distribution, and the poor efficacy at the target site.

The efficiency of DDSs can be affected by several parameters, including the dimensions of the NPs, their surface charge and their affinity toward the drug [6,7], and, not least, the interactions the NPs establish with the body and the tumor microenvironment [8].

DDSs based on NPs have been prepared using different materials, both inorganics and organics, as well as their combination [9,10,11]. Among them, polymers have been widely exploited for the NPs preparation; in fact, several non-toxic and biodegradable polymers—both natural and synthetic—have been employed for NPs synthesis aimed at DDSs [12,13]. The employed polymers included chitosan, poly(lactide-*co*-glycolide) (PLGA), poly(ethylene)glycol (PEG), and poly(caprolactone); moreover, their copolymers have often been considered [4,14]. 

Poly(L-lactide-*co*-caprolactone-*co*-glycolide) (PLCG) is a copolymer of polylactic acid, polycaprolactone, and polyglycolic acid. Compared to other polymers, for instance, PLGA, an advantage of PLCG is that, upon degradation, the release of acidic species is more contained [15]. This feature is beneficial in biomedical applications because a very acidic environment can lead to cytotoxicity. Nevertheless, relatively few studies have been published so far on the development of DDSs based on PLCG. PLCG NPs loaded with 5-fluorouracil were tested for anticancer activity [16], while in the work of Sanna et al., a block copolymer of poly(lactide-*co*-caprolactone-*co*-glycolide) was prepared and used for loading and delivering docetaxel to human prostate cancer cells [17]. More recently, porous poly(l-lactide-*co*-glycolide-*co*-ε-caprolactone) microparticles were developed as potential drug delivery vectors [18]. To the best of our knowledge, no more investigations on PLCG NPs and their performance for drug delivery were ever performed. 

In the preparation of NPs, a key role is played by the surfactant employed during the synthesis; the use of an appropriate surfactant, in fact, can lead to NPs with tailored dimensions and shape [19,20]. Moreover, according to the nature of the employed molecule, the surface charge of the NPs can also be different—this again can have a significant effect on the performance [21], as well as on the drug encapsulation efficiency [19]. 

In the present work, we report a systematic study on PLCG NPs as possible DDSs in cancer therapy. The NPs were prepared using two surfactants, polyvinyl alcohol (PVA) and dextran (DEX); a mixture of PVA/DEX was also considered. PVA is an FDA-approved synthetic polymer, generally used in food packaging and detergent compounds. It is a non-ionic surfactant already used in the preparation of PLGA nanoparticles [22], but also exploited as a biomedical material in the form of a hydrogel [23]. In addition, it undergoes enzymatic degradation under the action of microorganisms [23,24]. DEX, on the other hand, is one of the most used natural polymers: it is a polysaccharide widely employed in biomedicine due to its natural abundance and ease of modification, in addition to its excellent biocompatibility and biodegradability [25]. More recently, it has been used as a surfactant for inorganic nanoparticles and as a carrier system for oligonucleotides [26,27,28] To investigate the capability of the PLCG-based vectors to the host and release molecules with different chemical affinities, two model anti-cancer drugs were considered, the hydrophilic doxorubicin (DOXO) and the hydrophobic SN-38. The activity site of both DOXO and SN-38 is the nucleus and, despite their mechanism of action being different, the ultimate effect is to impair DNA replication, thus leading to the apoptosis of the tumor cells [29,30]. Both drugs are currently used in the treatment of some solid tumors, more commonly in combination with other chemotherapeutics. Nanoparticles-based delivery vectors of these drugs have been already proposed and, in some cases, successfully adopted in clinical use [31,32] The PLCG NPs were prepared by the emulsion evaporation method; a complete characterization of the particles prepared in different conditions was performed to determine the size, surface charge, thermal stability, morphology, etc. Tests to evaluate the drug encapsulation efficiency and the cytotoxicity were also performed.

## 2. Materials and Methods

### 2.1. Materials

All chemicals employed in this work were used as purchased, without further treatments and/or purification. Poly(L-lactide-*co*-caprolactone-*co*-glycolide) (PLCG, L-lactide 70%, average MW ~50,000), poly(vinyl alcohol) (PVA) 98% hydrolyzed, and dextran sulfate sodium salt (from Leuconostoc sp. DEX), cetyltrimethylammonium bromide (CTAB), doxorubicin (DOXO), 7-ethyl-10-hydroxycamptothecin (SN-38) dimethyl sulfoxide (DMSO), fetal bovine serum (FBS), trypsin, ethylenediamine tetraacetic acid (EDTA), 3-[4,5-dimethylthiazol-2-yl]-2,5-diphenyl tetrazolium bromide (MTT), and paraformaldehyde were purchased from Sigma-Aldrich. Acetone was obtained from J.T. Baker.

Dulbecco’s Modified Eagle’s medium (DMEM) was purchased by Gibco. Michigan Cancer Foundation-7 (MCF-7) breast carcinoma cells were obtained from ATCC.

### 2.2. Preparation of the NPs

To prepare the NPs, a PLCG solution (0.2 vol. %) was prepared in acetone. Solutions of either PVA, DEX, or PVA/DEX were prepared in water (0.1 vol. %).

In total, 2 mL of the surfactant solution was added drop-by-drop to the polymeric solution under stirring; the formation of an emulsion took place almost immediately. After stirring for 30 min, the solvent was removed by evaporation and the emulsion was left under vigorous stirring (1000 rpm on a vibrating Multi Reax shaker) for about 4 h. Then, 4 mL of ultrapure water was added. Successively, dialysis of the emulsion against water was carried out for 2 days to eliminate any excess polymer and/or surfactant, and the residue was finally filtered with a 5 µm filter to remove possible aggregates or large residues. The whole process was performed at room temperature.

The NPs concentration was determined by lyophilizing the suspension and accurately weighing the solid residue.

To load either DOXO or SN38 inside the NPs, the protocol used was the same as described above with a slight modification as follows. DOXO (concentration in the volume of the synthesis mixture equal to 51.5 µM) was dissolved in the aqueous phase, together with the surfactant. On the other hand, SN-38 was dissolved in DMSO and added to the PLCG solution in acetone (concentration in the volume of the synthesis mixture equal to 1.25 µM). These concentrations were chosen upon preliminary tests in order to obtain the best encapsulation performance with the lowest drug amounts.

### 2.3. Characterization of the NPs

The NPs were analyzed by thermogravimetric analysis (TGA), using an SDT Q600 equipment (TA Instrument, New Castle, DE, USA), with a heat ramp of 5 °C/min and an air flow rate of 100 mL/min. The NPs were lyophilized prior to the analysis. Fourier transform infrared (FTIR) spectra were taken with a Perkin Elmer Spectrum One Fourier Transform spectrophotometer (Waltham, MA, USA); each spectrum was acquired with 64 scans and a resolution of 4 cm^−1^. To acquire the spectra, all the samples were dissolved in water and drop-casting films were prepared directly on the ATR prism; spectra were collected after water evaporation.

The hydrodynamic dimensions of the NPs and their zeta potential values were measured with the Dynamic Light Scattering technique using a Zetasizer Nano ZS90 (Malvern Instruments Ltd., Malvern, UK); the NPs were suspended in ultrapure water.

Atomic Force Microscopy (AFM) and Transmission Electron Microscopy (TEM) were employed to study the morphology of the NPs. Non-Contact Atomic Force Microscope (NC-AFM) images were obtained at room temperature using an XE-100 microscope (Park Systems, Suwon, Korea). A drop of NPs suspension was deposited onto freshly cleaved mica and air-dried for several hours. PPP-NCHR probes (Nanosensors) with a nominal frequency resonance of 330 kHz were employed. AFM images were analyzed with the XEI 4.3 software. TEM analysis was performed with a JEOL Jem1011 microscope (Tokyo, Japan) operating at an accelerating voltage of 100 kV. The nanoparticle suspension was dropped onto carbon-coated copper grids that were left to dry under air overnight (O.N.) prior to being imaged.

Fluorimetric analysis was employed to assess the extent of the encapsulation of the drugs inside the NPs and their release. To this aim, fluorescence spectra were acquired using Cary Eclipse equipment (Agilent, Santa Clara, CA, USA). To determine the amount of the encapsulated drug, a calibration curve of free drugs in solution was prepared; for DOXO, the excitation wavelength was set at 488 nm, while fluorescence was detected at 595 nm; in the case of SN-38 the excitation wavelength was set at 370 nm, while the fluorescence was collected at 550 nm. The encapsulation efficiency (EE, %) was calculated according to the formula:(1)$$\mathrm{Encapsulation}\;\mathrm{Efficiency}\;\left(\%\right)=\frac{\mathrm{drug}\;\mathrm{added}\;\mathrm{in}\;\mathrm{solution}-\mathrm{non}\;\mathrm{encapsulated}\;\mathrm{drug}}{\mathrm{drug}\;\mathrm{added}\;\mathrm{in}\;\mathrm{solution}}\times 100$$

The amount of the non-encapsulated drug was measured after centrifugation and separation of the nanoparticles.

The loading capacity (LC, %) was also estimated according to the formula:(2)$$\mathrm{Loading}\;\mathrm{capacity}\;\left(\%\right)=\frac{\mathrm{mass}\;\mathrm{of}\;\mathrm{drug}\;\mathrm{encapsulated}}{\mathrm{mass}\;\mathrm{of}\;\mathrm{nanoparticle}}\times 100$$

To study the amount of drug released, the NPs suspension was diluted in phosphate-buffered solution with pH adjusted at either 4.5 or 7.4 value at 37 °C. The drug concentration was measured by fluorimetry at regular time intervals (0, 3, 6, 24, 48, and 72 h). The release efficiency was calculated according to the formula:(3)$$\mathrm{Release}\;\mathrm{Efficiency}\;\left(\%\right)=\frac{\mathrm{released}\;\mathrm{drug}\;\mathrm{concentration}}{\mathrm{encapsulated}\;\mathrm{drug}\;\mathrm{concentration}}\times 100$$

### 2.4. Cellular Studies

To assess the effectiveness of the prepared NPs as DDSs, cellular studies were performed on human cancer cells; more specifically, a human cell line of mammary carcinoma, Michigan Cancer Foundation 7 (MCF-7), was used. The cells were grown in a DMEM medium supplemented with 10% of fetal bovine serum (FBS), 2 mM glutamine, 100 IU/mL of penicillin and 100 μg/mL of streptomycin at 37 °C in a humidified atmosphere with 5% CO_2_.

#### 2.4.1. Cell Viability

Cell viability was evaluated by the 3-[4,5-dimethylthiazol-2-yl]-2,5-diphenyl tetrazolium bromide assay (MTT assay). Cells were seeded at a density of 2 × 10^4^ cells/well in three 96-well plates, to perform NP incubation at different time points, i.e., 24, 48, and 72 h in the case of DOXO and 24, 48 and 120 h in the case of SN-38. After 24 h incubation at 37 °C, the NPs were added at different concentrations (each point in triplicate) and cells were maintained in a humidified incubator with 5% CO_2_ at 37 °C. At the end of each predetermined incubation time, the culture medium was removed, the cells were washed twice with PBS, and 200 μL of fresh serum-free medium containing 2 mg/mL MTT were added to each well and incubated for 3 h at 37 °C. The MTT reagent was then removed and the formazan crystals were solubilized using dimethyl sulfoxide. The absorbance was read using a CLARIO star Plus microplate reader (570 nm). Experiments with no NPs were also performed as a negative control. The percentage of cell viability was determined according to the equation:(4)$$\mathrm{Viability}\;\left(\%\right)=\frac{\mathrm{Absorption}\;\mathrm{of}\;\mathrm{the}\;\mathrm{treated}\;\mathrm{sample}}{\mathrm{Absorption}\;\mathrm{of}\;\mathrm{the}\;\mathrm{control}\;\mathrm{sample}}\times 100$$

Based on these data, the IC_50_ values were calculated by plotting the viability versus the drug concentration (either DOXO or SN-38); experimental data were fitted with a linear regression, which was employed to calculate the concentration corresponding to a 50% viability value.

#### 2.4.2. Dichlorofluorescein Assay (DCF)

To measure the intracellular ROS levels, 2′,7′-dichlorofluorescein diacetate (DCFH-DA, Sigma-Aldrich) staining was used [33]. DCFH-DA is a stable, fluorogenic and non-polar compound that can readily diffuse into the cells and get deacetylated by intracellular esterases to a non-fluorescent 2′,7′-dichlorodihydrofluorescein (DCFH), which is later oxidized by intracellular ROS into highly fluorescent 2′,7′-dichlorofluorescein (DCF). The intensity of fluorescence is proportional to intracellular ROS levels. MCF-7 cells were seeded at a density of 1 × 10^5^ cells per well in 24 well plates and were allowed to attach overnight. Then, the medium was replaced with the fresh ones containing either the NPs (both empty and drug-loaded) or the free drugs. An H_2_O_2_ (1 mM) solution was used as the positive control. At the end of the incubation time, the spent medium was removed, and the cells were washed once with fresh DMEM, twice with 1× PBS and incubated with DCFH-DA (10 µM) for 30 min. After rinsing with PBS, representative fluorescent images were taken; the green fluorescent channel on an Evos m7000 fluorescence microscope (Thermo Fisher Scientific Inc., Waltham, MA USA) was used. Soon after, PBS was removed and the radioimmunoprecipitation assay (RIPA) buffer was added to each well. The collected cells were incubated at −80 °C for 20 min and then centrifuged at 21,130 *g* for 10 min at 4 °C. The supernatants were transferred to a black 96 well plate and the fluorescence intensity was measured, using a CLARIO star Plus microplate reader (BMG Labtech, Ortenberg, Germany) at an excitation wavelength of 485 nm and an emission wavelength of 530 nm. After fluorescence recording, 5 µL of supernatant were transferred to a clear 96 well plate containing 195 µL of the BCA protein assay solution to measure the protein concentration through the preparation of a calibration curve with BSA. The fluorescence intensity was normalized to the protein concentration.

### 2.5. Statistical Analysis

All data represent the average value of at least three independent experiments. Normally distributed data were compared with a two-tailed Student’s *t*-test. The bars in the graphs and tables represent the mean ± S.D. values.

## 3. Results and Discussion

### 3.1. Synthesis and Structural Study of the Different PLCG NPs

As reported in the experimental section, PLCG NPs were prepared using PVA and DEX as surfactants. The combination of the two surfactants, PVA and DEX, was also considered to evaluate how the features of each polymer affect the properties of the resulting nanoparticles. The general scheme of the work and the components used for the preparation of the nanoparticles are reported in Figure 1. The nanoparticles were formed by the slow addition of the aqueous solution of the surfactants to the acetone solution of the polymer, followed by solvent evaporation.

To have better knowledge and understanding of the different systems, a preliminary characterization was performed.

The different NPs were analyzed by FT-IR spectroscopy and the free polymer (PLCG) and the surfactants were also analyzed for comparison; the spectra are reported in Figure 1a for the free compounds and 1b for the NPs.

The free PLCG spectrum (Figure 1a) shows signals in agreement with those given in the literature [34,35]. More specifically, the observed peaks belong to the stretching modes of the C–O groups (1180 and 1085 cm^−1^) and to the C=O group (1752 cm^−1^); the CH_3_, CH_2_, and CH stretching mode (3000−2800 cm^−1^) and the deformation vibrations of CH_3_ (polylactide segment) at 1352 and 1378 cm^−1^ can also be observed, together with the wagging modes of CH_2_ (polyglycolide segment) at 1418 and 1452 cm^−1^. Moreover, the compound was anhydrous, since no absorption bands between 3600 and 3400 cm^–1^ were observed.

The spectra of free surfactants (Figure 1a) show the typical vibrations of these compounds, as reported in the literature for PVA and DEX—C–Hx, C–O, and O–H vibrations [36,37].

In Figure 1b the spectra corresponding to the PLCG nanoparticles are reported; all of them are almost totally superimposable with that of free polymer; some differences, however, can be observed: in fact, the C=O stretching peak is 7 cm^−1^ shifted towards higher frequencies (from 1752 to 1759 cm^−1^) (see the asterisk in the graph); moreover, the intensity ratio between the C–O/C=O bands (1084 and 1759 cm^−1^, respectively) decreases (see the two asterisks). These differences, although small, may be ascribed to some kind of interaction between the surfactants and PLCG, e.g., between the alcoholic groups of the former and the C=O group of the latter.

To understand the stability of the systems and the possible interactions between the polymer and the surfactants, TGA measurements were carried out—see Figure 2. In the tests, the behavior of the NPs with the surfactants (Figure 2a) is compared with that of the free components (Figure 2c); together with the TGA curves, their first derivatives were also considered (Figure 2b,d) to better evidence the successive steps in the weight loss. Considering the free PLCG, it can be seen that a complete weight loss is observed for temperatures up to 600 °C; indeed, the main weight loss can be observed for 300 < T < 400 °C, in agreement with the literature data [15]. A smaller peak is present just around 450 °C.

However, the thermal behavior of PLCG changes when it is in the form of NPs depending on the surfactant(s) used in the NPs preparation. From Figure 2a,b, it can be seen that thermal degradation takes place over several different steps. For all systems, a signal can be observed for T < 50–70 °C, which corresponds to the desorption of water molecules and whose intensity depends on the amount of solvent desorbed. 

The peaks corresponding to the degradation of the PLCG NPs were observed at different temperatures, depending on the surfactants. For both DEX and PVA/DEX, almost no shift was observed in comparison with the free polymer; in fact, signals were observed at about 350 and 450 °C. This indicates that the interactions taking place due to the presence of these surfactants do not affect significantly the stability of the polymer. With PVA, on the contrary, no peak was observed for T < 400 °C, but the decomposition of the polymer takes place at about 430 and 450–460 °C, indicating higher thermal stability. 

More peaks can be detected for T < 300 °C, likely corresponding to the degradation of the surfactants. For PVA it can be seen that the degradation of the molecule takes place in two steps, at about 170 and 280 °C; again, differences can be seen with the free surfactant where no signal is present at 170 °C. For the DEX curve, on the other hand, a large peak can be observed for T < 200 °C overlapping a much sharper one. The same behavior is seen in the NPs prepared with DEX/PVA; for free DEX, on the contrary, the degradation takes place only for T > 200 °C.

All these differences show that the nature of the surfactant(s) plays a key role in the thermal stability of the PLCG polymer; these features can potentially affect the performance of the NPs for drug delivery [38]. 

To evaluate the loading and delivery ability of these two drugs containing nanoparticles, the hydrophilic DOXO and the hydrophobic SN-38 were encapsulated. Figure 2 reports the chemical structure of the two molecules. 

Table 1 shows the DLS data (average hydrodynamic size, polydispersity index PDI, and particle charge) for the empty NPs and those loaded with DOXO and SN-38. The average size of the empty NPs varies from 175 nm, in the case of PVA-coated NPs, to 222 nm, in the case of the DEX. When they are loaded with the drugs, the average hydrodynamic diameter shifts to larger values, i.e., around 310 nm in the case of SN-38-loaded NPs and 370 nm with DOXO. The extent of the difference, however, varies depending on the system considered. As a general comment, for all systems, PDI values are relatively small, indicating that the formulations have a homogeneous size distribution. Appendix A (see Appendix A) shows as an example the curves of the size distribution of the NPs prepared with PVA/DEX, empty and loaded with the two drugs. 

To test the stability of the NPs over time, their size was measured for 21 days while keeping the NPs at 37 °C—see Figure 3. It can be seen that for all systems, regardless of the surfactant used or the drug loaded (Figure 3a for DOXO and Figure 3b for SN-38), the size did not change remarkably; in fact, only small variations were observed within the experimental error and with no obvious trend. Regarding the surface charge, negative values were measured for all the NPs, as already reported [26,39]. 

### 3.2. Morphology of the NPs

TEM and AFM characterizations were performed to provide an exhaustive analysis of the morphology of the NPs. 

Figure 4 shows TEM micrographs for the NPs prepared with the various surfactants. The empty NPs (Figure 4a,d,g), regardless of the type of surfactant, showed an average size of about 130 ± 30 nm. On the other hand, the encapsulation of either DOXO (Figure 4b,e,h) or SN-38 (Figure 4c,f,i) led to larger and more uniform particles (about 300 nm) that display rough contours and higher contrast under the electron beam. By comparing the TEM sizes of the nanoparticles with those obtained by means of DLS analysis, it can be observed that the size values are quite similar. 

AFM imaging of the empty NPs, as reported in Appendix A, confirmed the data obtained by TEM and DLS analysis: the empty nanoparticles, in fact, do not display uniform size distribution, and those coated with DEX look slightly larger than the other two types. 

### 3.3. Optical Analysis and Encapsulation of the Drugs

To determine the amount of drug encapsulated in the PLCG NPs, fluorimetric spectroscopy was used; Appendix A shows the spectra of free and encapsulated DOXO and SN-38. Considering the DOXO (Appendix A), the spectrum of the free drug (black line) showed the same qualitative behavior reported in the literature with a maximum peak at about 592 nm [40]. The green line represents the spectrum of the DOXO-loaded NPs prepared using PVA/DEX as surfactants; it can be seen that a peak at the same wavelength is present, although with lower intensity, representing the concentration of the encapsulated drug. NPs prepared with the two surfactants individually, i.e., either PVA or DEX, showed similar features (data not shown). 

The EE (%) values for each type of NP system are reported in Table 2; they were estimated by measuring the amount of drug not encapsulated in comparison to the DOXO feeding solution. These data confirmed that the type of surfactant and their interaction with the drug affect the DOXO encapsulation efficiency. Indeed, the lowest efficiency was measured with PVA (72%) while the highest was with DEX (87%). An intermediate value was registered for the system with both surfactants (82%). The highest performance of DEX-coated NPs might be explained by the hydrophilicity of DOXO and its tendency to accommodate hydrophilic environments [41]. The LC% values were also calculated and are similar to other polymeric nanoparticles loaded with DOXO [38]. 

Analysis of the fluorescence spectra of SN-38 (Appendix A) showed that the solubilization medium and the environment affect the fluorescence emission of the molecule, with the free drug dissolved in water (black curve) displaying the maximum photoluminescence at about 555 nm. The curves of the fluorescence emission of the free drug dissolved in DMSO (red curve) showed two peaks (at 406 and 426 nm), while once it was hosted in the nanoparticle matrix the molecules showed one emission peak at 424 nm. This behavior differs from what was observed with the DOXO and reflects the different affinity of the two molecules for the aqueous environment.

As shown in Table 2, the lowest encapsulation efficiency (~56%) was obtained with the surfactant mixture, while the highest was in the nanoparticles prepared with DEX (~87%). An intermediate value was estimated in the case of PVA-coated NP (~73%). The encapsulation efficiency of a drug inside a polymeric nanoparticle can be affected by many different features, one key element being the affinity between the polymer and the drug [42]. Interestingly, these data evidenced that the surfactant can also be a determinant; in fact, our results show that DEX was the most efficient molecule for both the hydrophilic and hydrophobic drugs. The literature data also reported the significant effect of the surfactant; Shkodra-Pula B et al. (2019) indeed reported that a hydrolyzed PVA surfactant is more effective than poloxamers and polysorbates with PLGA nanoparticles in terms of both colloidal stability and drug entrapment percentage.

### 3.4. Cellular Studies

In vitro experiments were performed on breast cancer cells, namely MCF-7, to assess the drug delivery efficacy of the different NPs systems. Preliminary viability studies were carried out to determine the biocompatibility of the polymeric matrices combined with the different surfactants through a standard MTT test.

Obviously, the biocompatibility of the nanoparticle components represents the first requisite of a drug delivery system; in fact, potential surfactants, which could provide optimal features to the NPs in terms of size, stability and EE, may be excluded if the resulting systems are not biocompatible. Indeed, in the first steps of this work, an additional surfactant was used to prepare the PLCG nanoparticles, cetyltrimethylammonium bromide (CTAB). Appendix A shows AFM images of the NPs, and it can be seen the CTAB-coated NPs were very uniform and exhibited small size. Unfortunately, however, they were very toxic, as the viability of cancer cells incubated with the empty NPs reached values of about 10% at 120 h (see Appendix A with MTT test data, grey curve). These results are in agreement with previously published studies indicating that NPs prepared with this surfactant may show cytotoxic behavior [43]. Considering these results, no more tests were performed with the CTAB-based NPs.

Considering the other surfactants, they showed to be more suitable as DDSs, as it can be observed from the data of the MTT viability assay (Appendix A). The cell viability for the empty NPs (grey curves), in fact, were shown to be above 75% for all the NP concentration ranges used (ranging from 24 to 200 µg/mL, see also Appendix A) and for up to 5 days of incubation. The interval of drug amounts, either free or encapsulated, to be administered to the cells was selected upon evaluation of the cell line sensitivity to the two drugs, as reported in the drug database (https://www.cancerrxgene.org/, accessed on 1 March 2022). 

The results of the DOXO-loaded particles are reported in Appendix A (red curves); cell viability was compared to that of cells exposed to the free drug (black curves) in the concentration range from 1 to 4 µM. Different incubation times were sampled for up to 3 days. It can be seen that, apart from a couple of exceptions, there is no appreciable difference between the loaded NPs and the free drug for lower incubation times; at 72 h, however, some DDSs were shown to be more effective than the free DOXO. To perform a proper comparison, the IC_50_ values were calculated for this longer time; data are shown in Table 3.

It can be seen that the NPs prepared with DEX are the most effective; indeed the IC_50_ value is 2.2 µM, about 35% lower than the free DOXO (3.4 µM). The mixture of DEX and PVA was also shown to be valid, as a comparable IC_50_ value was registered (2.5 µM). The difference between the IC_50_ values of the MCF-7 cells incubated with free DOXO and with either DEX or PVA/DEX NPs loaded with DOXO is statistically significant (*p* < 0.01). On the other hand, NPs prepared using just PVA did not show a good performance, as the IC_50_ value was higher than that of the free drug (5.6 and 3.4 µM, respectively). 

For SN-38, free drug concentrations between 100 and 500 nM and incubation times up to 5 days were considered; the results of the MTT test are shown in Appendix A. The results of the viability assay indicate that none of the NPs formulations were more effective than the free SN-38 for the whole length of the incubation. This trend was confirmed also by the IC_50_ values calculated for an incubation time of 5 days (see Table 3); indeed the values for PVA- and PVA/DEX-coated NPs are 500 and 400 nM, respectively, about twice the value obtained with the free drug (200 nM). The performance of the NPs prepared with DEX, on the other hand, was comparable to that of the free SN-38, with an IC_50_ value of 300 nM. The better results obtained with DEX for SN-38 are in agreement with what was observed with DOXO.

Moreover, upon determination of the loading capability of the nanosystems and the number of nanoparticles administered, the IC_50_ of the cells administered with either empty or loaded NPs was also estimated and compared (see Appendix A). The values are reported as the amount of NPs (mass in µg) needed to kill 50% of cells. The data confirm the good biocompatibility of the nanoparticles, displaying those coated with PVA with the highest values (454 µg after 120 h), while those with DEX with the lowest (349 µg after 120 h). Reasonably, the loading of the drugs significantly reduced (from 4 to 9 times) the amount of NP needed to reach IC_50_.

As a general comment, the IC_50_ value of the MCF-7 cells treated with free DOXO and SN-38 are quite different, being that of DOXO 10 times higher than SN-38. This response should be related to the metabolization of the drug upon internalization and the development of resistance mechanisms. In this sense, the use of NP-based carriers may mitigate this activity bypassing the classic internalization route of the free compound and facilitating the entry into the cytosol.

It is known that anticancer drugs, in addition to their antiproliferative effect, induce directly or indirectly oxidative stress [44,45]. Thus, to evaluate the production of reactive oxygen species (ROS) upon internalization and metabolization of the two drugs, the DCF assay was performed. MCF-7 cells were incubated with the three types of NPs at a drug concentration below the IC_50_ value (1 µM and 100 nM in the case of DOXO and SN-38, respectively). The results are presented in Figure 5a; they show that both drug treatments cause an increase in the intracellular ROS level, and the effect is more remarkable in the case of DOXO. Indeed, the cells administered with PVA/DEX and DEX NPs loaded with DOXO display an increase of the fluorescent signal of about 4 and 4.5 times, respectively, compared to control samples. In the case of the free drug and of the NPs prepared with PVA, a smaller increase in the fluorescence is observed (up to 3.4). Moreover, the images in Figure 6 show brightly fluorescent rounded cells in the samples incubated with either free or encapsulated DOXO. These results are in agreement with those of the viability assay.

For the SN-38, on the contrary, data from Figure 5b indicate that the free drug induces a slight increase in intracellular ROS, while the encapsulated drug boosts the fluorescence intensity more—increases of 2.4, 2.8 and 3.6 times were observed for DEX, PVA/DEX and PVA NPs, respectively (compared to the control). These data seem to indicate that SN-38-loaded NPs are more efficient mediators of ROS-related response than the free SN-38; this is not in agreement with the results of the cytotoxic assay. It is likely that multiple intracellular processes may interplay with the drug-based response. The images in Figure 6a,b show few fluorescent and detaching cells over the imaged area, suggesting a limited ROS-mediated stress response in cells incubated with both free and loaded SN-38. 

To elucidate the mechanism behind the cell viability, the kinetics of DOXO and SN-38 release were investigated; two different pH conditions were chosen—4.5 and 7.4—to reproduce the conditions of the cancer microenvironment and physiological environment, respectively (data shown in Figure 7). 

In the case of DOXO (Figure 7a,b) for all NPs and in both pH conditions all curves show an initial burst in the first 8 h, and then a slow increase is observed at each time point, reaching a maximum after 72 h. Longer times were not studied as the curves showed a stable behavior, i.e., a plateau with no further increase.

The lowest release was observed for the NPs prepared using PVA; indeed, DOXO release reaches a maximum of about 34% and 30% at pH 4.5 and 7, respectively, after 72 h, corresponding to a concentration of about 10.8 and 9.6 µM. The highest release was detected for the DEX system that reaching 48% and 42% at pH 4.5 and 7, respectively, after 3 days of incubation. An intermediate release percentage was observed for the nanoparticles coated with PVA/DEX. These trends are in agreement with the results of the viability assay suggesting that PVA is less effective as a delivery vector due to a poor release in the period under investigation. Notably, the weak cytotoxic effect of the DOXO-loaded NPs observed for shorter incubation times might be due to the amount of drug released at that time.

Considering the release curves of the SN-38 loaded nanoparticles (Figure 7c,d), interestingly, a very limited release of the drug at both pH values in the case of PVA- and PVA/DEX-coated NPs can be observed. Indeed, the two curves exhibit a flat profile reaching a maximum of 5% and 6.5% release after 5 days, respectively. On the other hand, the DEX-coated nanoparticles show a time-dependent behavior with 25% release after 24 h and a maximum release equal to 50% at pH 7 and almost 40% at acidic pH after 5 days. These data indicate that the interactions SN-38 establishes with PVA are stronger than those with DEX, thus hindering the release of the drug from the nanoparticles prepared with PVA, either alone or combined with DEX. Such limited SN-38 release was not observed with other NPs prepared using PVA; the literature data, for instance, report that SN-38 releases up to 60% for PLGA-based NPs [46]. On the other hand, in another study, PEG-PLA nanoparticles showed release kinetics inversely dependent on the length of the PLA chain, with the shorter polymer being the best performing (almost 40% release) up to 5 days [47]. Our results may indicate that both the polymer and the surfactant, and the combination of the two, affect the interaction with the drug and, hence, its release. This complex topic may need a more detailed investigation.

The SN-38 release results are in agreement with the cell viability data, as only NPs coated with DEX exert a cytotoxic activity comparable to that of the free drug.

Overall, it can be stated that DEX is more efficient as a surfactant to prepare NPs for drug delivery of both hydrophilic and hydrophobic drugs; moreover, its combination with PVA does not lead to any improvement in performance. The PLCG-based DDSs are more effective when a hydrophilic drug is employed.

## 4. Conclusions

In this study, poly(l-lactide-*co*-caprolactone-*co*-glycolide) was used as a polymeric backbone to prepare organic NPs coated with different surfactants (DEX, PVA, and a mixture of the two) via the solvent evaporation method. The nature of the surfactant influenced both size and thermal stability of the NPs, as well as the capability to host two chemotherapeutics, the hydrophilic DOXO, and the hydrophobic SN-38. NPs coated with DEX exhibited the highest encapsulation efficiency with both drugs (up to 87%) and the best release profiles under the set conditions. Cell viability assays performed on the MCF-7 cell line showed that the empty NPs were highly biocompatible and the DEX-coated particles were the most effective as cytotoxic agents. Indeed, they reached IC_50_ values lower than the free drug in the case of DOXO and close to that of the free compound in the case of SN-38. On the other hand, the low cytotoxicity of the NPs coated with PVA and PVA/DEX NPs and loaded with SN-38 is likely associated with the limited drug release kinetics. The latter might depend on the strong interactions between the surfactant and the hydrophobic drug. Remarkably, the average size of the loaded NPs is larger than 300 nm, a diameter that exceeds the optimal range for in vivo application of nanoparticles-based carriers. Beyond this limitation, the interest in the development of nanocarriers for topical delivery of active compounds is gaining more attention. In this sense, DEX-coated PLCG NPs look like promising delivery tools and deserve additional investigation. Moreover, their combination with other surfactants will also be studied to improve the delivery performance of hydrophobic compounds.

## Data Availability

Data sharing is not applicable to this article.

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
