# Peer review of "Poly(l-lactide-co-caprolactone-co-glycolide)-Based Nanoparticles as Delivery Platform: Effect of the Surfactants on Characteristics and Delivery Efficiency"

_nanomaterials, 2022, doi:10.3390/nano12091550_

Round 1
Reviewer 1 Report
This work is devoted to effect of the surfactants on characteristics and delivery efficiency of PLCG-based nanoparticles. Authors prepared polymeric nanoparticles made of the copolymer Poly(L-lactide-co-caprolactone-co-glycolide) using two surfactants, polyvinyl alcohol (PVA) and dextran (DEX); a mixture of PVA/DEX was also considered. The PLCG NPs were prepared by the emulsion evaporation method; a complete characterisation of the particles prepared in different conditions was performed to determine size, surface charge, thermal stability, morphology, etc. Tests to evaluate the drug encapsulation efficiency and the cytotoxicity were also performed. The work is of interest because these developed nanocarriers are ideal candidates to deliver hydrophilic drugs. Taking into account the mentioned below notes, I think that the article looks like a short communication and may be published after major revision.
Notes:
- The general scheme of the present study should be presented in the text for better understanding of possible use PLCG-based nanoparticles as possible DDSs in cancer therapy. Also the structures of studied chemotherapeutic drugs doxorubicin and SN-38 should be shown on this scheme.
- Why IC50 values of DOXO and SN-38 for the studied delivery systems are so different? The discussion about it should be added in the text.
- From what point of view the authors choose the hydrophilic doxorubicin and the hydrophobic SN-38 as chemotherapeutic drugs? It is clear that the hydrophobic drug was chosen for the possibility of encapsulation in order to improve the ability to dissolve in aqueous media. But why did the authors still investigate a hydrophilic drug in their work?
Author Response
We thank the reviewers for the detailed reading and the suggested modifications. According to these the quality and the content of the manuscript will result ultimately improved.
Below a detailed response to the points raised by the reviewers is reported. The comments of the reviewers are in bold while our replied are in blue. In the manuscript the edited parts are in red, for the reviewers to see the changes more easily.
Reviewer 1
- The general scheme of the present study should be presented in the text for better understanding of possible use PLCG-based nanoparticles as possible DDSs in cancer therapy. Also the structures of studied chemotherapeutic drugs doxorubicin and SN-38 should be shown on this scheme.
The following scheme was added into the revised manuscript (page 8):
Scheme 1. Structure of the two drugs loaded into the nanoparticles, (a) DOXO and (b) SN-38.
- Why IC50 values of DOXO and SN-38 for the studied delivery systems are so different? The discussion about it should be added in the text.
As reported in literature and in a previous work by some of us (Zacheo et al., Frontiers in Bioengineering and Biotechnology 2020, 8, doi:10.3389/fbioe.2020.00690) the two drugs display different pharmacokinetic profiles and efficacy, both in vitro and in vivo and consequently different IC50. In addition, it is likely that the cells developed resistance to drug treatments upon prolonged subculturing. In this sense, the use of NP-based carrier may mitigate this activity bypassing the classic internalization route of the free compound and facilitating the entrance into the cytosol. A sentence has been added in the revised text.
- From what point of view the authors choose the hydrophilic doxorubicin and the hydrophobic SN-38 as chemotherapeutic drugs? It is clear that the hydrophobic drug was chosen for the possibility of encapsulation in order to improve the ability to dissolve in aqueous media. But why did the authors still investigate a hydrophilic drug in their work?
The encapsulation of a hydrophilic drug that per se can spontaneously enter the cell allows on one hand to protect the drug from degradation, sequestration and inactivation upon injection, and on the others to overcome cellular resistance mechanisms that could limit the drug entrance or activity bypassing the natural cellular defences.
Reviewer 2 Report
The subject of the manuscript “PLCG-based nanoparticles as delivery platform: effect of the surfactants on characteristics and delivery efficiency” by Magda M. Rebanda et al. is focused on the fabrication by the solvent evaporation method of copolymer Poly(L-lactide-co-caprolactone-co-glycolide) nanoparticles using polyvinyl alcohol and dextran as surfactants. A complete characterization of the particles prepared in different conditions was performed to determine size, surface charge, thermal stability, and morphology. Tests to evaluate the drug encapsulation efficiency and the cytotoxicity were also accomplished.
The idea of the manuscript is interesting. It can be accepted for publication after the authors will properly address all the raised queries (in the order they appear in the manuscript):
- The authors should not use abbreviations in the title.
- What was the applied speed for stirring?
- Did the authors perform their experiments at room temperature?
- A subsection specially devoted to statistical analysis should be also considered by the authors.
- Does the distribution of the particles have an effect on the delivery efficiency?
- Figures & Tables – all the abbreviations used should be described in the full form under the footnote for tables and in the notes following figure caption for figures. This is important as each figure & table should be independently understandable without reference to the text.
- Are the results presented in Fig. 5a,b statistically relevant or not?
- Did the authors study what is happening with the DOXO samples after 72h? (reference to Fig. 6a,b).
- At least one possible application of the fabricated nanoparticles should be indicated by the authors.
Author Response
Response to the reviewers
We thank the reviewers for the detailed reading and the suggested modifications. According to these the quality and the content of the manuscript will result ultimately improved.
Below a detailed response to the points raised by the reviewers is reported. The comments of the reviewers are in bold while our replied are in blue. In the manuscript the edited parts are in red, for the reviewers to see the changes more easily.
Reviewer 2
- The authors should not use abbreviations in the title.
The abbreviation was removed and replaced with the full name of the employed polymer.
- What was the applied speed for stirring?
The stirring speed was set at 1000 rpm. This information has been included in the experimental section 2.2.
- Did the authors perform their experiments at room temperature?
Yes, a sentence was added specifying this in the experimental section 2.2.
- A subsection specially devoted to statistical analysis should be also considered by the authors.
A section has been included in the experimental part (2.4). The statistical study of the data were included in the discussion of Table 3 and Figure 5.
2.4. Statistical analysis
All data represent the average value of at least three independent experiments. Normally distributed data were compared with a two-tailed Student’s t-test. The bars in graphs and Tables represent mean ± S.D. values.
- Does the distribution of the particles have an effect on the delivery efficiency?
Several studies and reviews dealt with the size issue since in drug delivery “the size matters”. It has been demonstrated that the cellular uptake and the pharmacokinetics depend on the carrier size (Nanomedicine 2016 Mar; 11(6): 673-692; The effect of nanoparticle size on in vivo pharmacokinetics and cellular interaction). In this sense, the optimal carrier has to be designed for each specific application. This aspect becomes more critical moving from cellular studies to in vivo studies.
The average hydrodynamic diameter of the drug-loaded NPs is around 340 nm. We are aware that this value is quite high, especially for in vivo systemic injection. Nevertheless, topical use of polymeric nano- and microparticles for local delivery of drugs though the skin is recently emerging. In this sense the use of systems with size larger than 200 nm would not represent a limitation.
However, the main focus of this study was the analysis of the effects of the surfactants over the nanoparticles features and drug encapsulation.
A sentence has been included in the revised manuscript at page 8 to highlight this point.
- Figures & Tables – all the abbreviations used should be described in the full form under the footnote for tables and in the notes following figure caption for figures. This is important as each figure & table should be independently understandable without reference to the text.
We thought that, adding the full form in every caption/footnote, there would have been too many repetitions in the text. To address the point raised by the reviewer, a list of all the abbreviations was added in the SI.
- Are the results presented in Fig. 5a,b statistically relevant or not?
We thank the referee for this suggestion. The statistical analysis of the data from the cellular tests has been performed and included in the revised manuscript.
Figure 5. DCF assay (reported as the fluorescence intensity/protein content, normalized to the control sample) performed with MCF-7 cells incubated with the NPs coated with the three types of surfactants and loaded with either (a) DOXO (for 72 hours) or (b) SN-38 (for 120 hours). The cells were administered with a drug concentration equal to 1 µM and 100 nM in the case of DOXO and SN-38, respectively. H2O2 treatment (1 mM) for 1 hour was used as positive control. (*) and (**) indicate statistical significance with p< 0.05 and 0.01, respectively.
- Did the authors study what is happening with the DOXO samples after 72h? (reference to Fig. 6a,b).
A sentence was added explaining why longer times were not studied.
- At least one possible application of the fabricated nanoparticles should be indicated by the authors.
In this work we developed the nanoparticles for the preliminary investigation of their use as delivery systems. The analysis of their size does not suggest their application in vivo as drug vector upon systemic injection in cancer therapy. On the other hand, the characterization of their performances evidenced that the nanoparticles are able to load and release hydrophobic and hydrophilic drugs, whose release is quite slow over time. These features in addition to the high biocompatibility would make this system a good candidate as drug vector for topical delivery of active compounds through the skin, as already demonstrated for PLGA nanoparticles for transdermal delivery of diclofenac. A sentence has been included in the Conclusion.

Reviewer 3 Report
‘PLCG-based nanoparticles as delivery platform: effect of the surfactants on characteristics and delivery efficiency’. I would suggest a slight modification in title. Instead of writing ‘delivery efficiency’, it is better to write ‘drug release’. The authors have performed the in vitro cytotoxicity studies and it predominantly involves ‘drug release’. ‘delivery efficiency’ would be suitable if in vivo studies also show promise.
The abstract is a bit longer (~220 words). I suggest it to be reduced to 200 words.
This is a study involving the role of surfactants. I have doubts over the novelty of the study.
Some of the studies performed by the authors are a dated. New techniques should be used for the characterization. FTIR could be replaced with NMR. Particle size analysis can be done using a coulter counter.
The quality of fluorescence images is weak. I would suggest the authors to include better quality images.
The authors did not mention about the statistical analysis. I could not see the p values etc. Analysis is incomplete without statistics.
I would recommend the authors to perform an in vivo tumor model study to assess the efficacy of delivery platform.
The number of references can also be slightly reduced.
Author Response
Response to the reviewers
We thank the reviewers for the detailed reading and the suggested modifications. According to these the quality and the content of the manuscript will result ultimately improved.
Below a detailed response to the points raised by the reviewers is reported. The comments of the reviewers are in bold while our replied are in blue. In the manuscript the edited parts are in red, for the reviewers to see the changes more easily.
Reviewer 3
The abstract is a bit longer (~220 words). I suggest it to be reduced to 200 words.
The abstract was reduced to 200 words.
This is a study involving the role of surfactants. I have doubts over the novelty of the study.
We are aware that in literature several studies describe the role of surfactants in the delivery performance of polymeric nanoparticles. The novelty of the present work is the combination of the PLCG copolymer with the two surfactants, the surfactant mixture and the study of their mutual interaction. As stressed in the Introduction, so far it has been poorly investigated in delivery studies.
Some of the studies performed by the authors are a dated. New techniques should be used for the characterization. FTIR could be replaced with NMR. Particle size analysis can be done using a coulter counter.
We agree with the reviewer comment. NMR would provide additional information about the structure of the NPs. At the moment, this analysis cannot be performed in our Institute. We rely on the reviewer understanding.
The quality of fluorescence images is weak. I would suggest the authors to include better quality images.
We agree with the referee: the images have been improved and the two channels (fluorescent and bright field) have been separated. In addition, in the revised manuscript the images are presented as a new Figure to provide larger panels.
Figure 6. Representative images of DCF staining in MCF-7 cells incubated with either the free drugs or with the PVA/DEX NPs, both empty and loaded. Panels (a) and (b) correspond to the fluorescence channel and to the bright field, respectively. The scale bar corresponds to 100 µm.
The authors did not mention about the statistical analysis. I could not see the p values etc. Analysis is incomplete without statistics.
We thank the referee for this suggestion. The statistical analysis of the cellular data has been performed and included in the revised manuscript. Here, we include the statistical analysis of Figure 5. It has also been performed on IC50 data.
Figure 5. DCF assay (reported as the fluorescence intensity/protein content, normalized to the control sample) performed with MCF-7 cells incubated with the NPs coated with the three types of surfactants and loaded with either (a) DOXO (for 72 hours) or (b) SN-38 (for 120 hours). The cells were administered with a drug concentration equal to 1 µM and 100 nM in the case of DOXO and SN-38, respectively. H2O2 treatment (1 mM) for 1 hour was used as positive control. (*) and (**) indicate statistical significance with p< 0.05 and 0.01, respectively.
I would recommend the authors to perform an in vivo tumor model study to assess the efficacy of delivery platform.
In vivo tumor model study were beyond the scope of this work, whose objective was the optimisation of the nanoparticles as anti-cancer drug carriers, both in terms of choice of the surfactants and of the kind of drug. Based on the evidences collected and upon investigation of the morphostructural features of the nanoparticles and of the delivery performances, we believe that studying their use as drug carriers for the topical (not systemic) delivery of active compounds would become very interesting. Therefore, this type of investigations will be performed on the DEX-based nanoparticles using DOXO. A sentence stating this was added in the conclusion.
The number of references can also be slightly reduced.
In the revised manuscript two references have been removed.

Round 2
Reviewer 1 Report
Authors have corrected almost all comments in the paper. But authors did not add the general scheme of the present study that, in my opinion, should be presented in the text for better understanding of possible use PLCG-based nanoparticles as possible DDSs in cancer therapy. I hope these corrections improved the paper and the revised version corresponds to high standards of Nanomaterials. After careful consideration, I think that this manuscript may be published after correction mentioned above.
Author Response
We agree with the referee: the addition of a scheme will help the reader to define the main points of the work.
Therefore, we have prepared and included the following Scheme in the revised version of the manuscript.
Sheme 1. Sketch of the polymeric components of the NPs and presentation of the steps of the work.

Reviewer 3 Report
The authors have revised the manuscript as per most of the suggestions and it has improved a lot. I think it is suitable for publication now.
Author Response
We thanks the referee for his/her positive feedback.